# Out of Distribution Detection and Adversarial Attacks on Deep Neural Networks for Robust Medical Image Analysis

**Anisie Uwimana** [1]   **Ransalu Senanayake** [2]

## Abstract

Deep learning models have become a popular choice for medical image analysis. However, the poor generalization performance of deep learning models limits them from being deployed in the real world as robustness is critical for medical applications. For instance, the state-of-the-art Convolutional Neural Networks (CNNs) fail to detect adversarial samples or samples drawn statistically far away from the training distribution. In this work, we experimentally evaluate the robustness of a Mahalanobis distance-based confidence score, a simple yet effective method for detecting abnormal input samples, in classifying malaria parasitized cells and uninfected cells. Results indicated that the Mahalanobis confidence score detector exhibits improved performance and robustness of deep learning models, and achieves state-of-the-art performance on both *out-of-distribution (OOD)* and *adversarial* samples.

## 1. Introduction

Deep learning is increasingly making its way into ground-breaking technologies that have high-value applications in the real-world clinical environment. Innovative medical imaging applications and diagnostics are among the most exciting use cases. One such application is developing microscopy-based malaria diagnosis procedures (Ravendran et al., 2015; Silva et al., 2013; Yang et al., 2019). Malaria is a deadly mosquito-borne disease infecting around 300 million people annually (World Health Organization). Since it is mostly prevalent in low-income countries, developing semi-automated microscopy techniques, as alternatives to polymerase chain reaction (PCR) tests and rapid diagnostic tests

[1]African Institute for Mathematical Sciences (AIMS), Rwanda, Kigali [2]Durand Building, 496 Lomita Mall, Stanford University, Stanford, CA 94305. Correspondence to: Anisie Uwimana <auwimana@aimsammi.org>.

*Accepted by the ICML 2021 workshop on A Blessing in Disguise: The Prospects and Perils of Adversarial Machine Learning.* Copyright 2021 by the author(s).

(RDT), is a low-cost and reliable solution (Wongsrichanalai et al., 2007).

### 1.1. Applications of deep learning in medical diagnosis

Esteva et al. (2017) developed a convolutional neural network (CNN) model that was trained on $130,000$ clinical images of skin pathologies to detect cancer. The proposed model achieves performance on par with all tested experts, demonstrating an artificial intelligence model capable of classifying skin cancer with a level of competence comparable to dermatologists. In 2018, another research study showed that a convolutional neural network trained to analyze dermatology images identified melanoma with ten percent more specificity than human clinicians (Haenssle et al., 2018). Another algorithm trained on $42,000$ chest CT scans outperformed expert radiologists in detecting lung cancers (Ardila et al., 2019). It was able to find malignant lung modes $5\text{-}9.5\%$ more often than human specialists. Recently, a CNN model designed to predict malignancy and identify 134 skin disorders (Cho et al., 2020). The proposed algorithm is capable of distinguishing, at the human expert level, melanoma from birthmarks. There have also been various studies on assessing the uncertainty and robustness in medical data (Senanayake et al., 2016; Laves et al., 2020; Asgharnezhad et al., 2020).

### 1.2. Applications of deep learning for malaria diagnosis

Various computer vision algorithms have been used for malaria diagnosis (Ravendran et al., 2015). Deep learning algorithms are recently being used increasingly by researchers especially for malaria detection because of its applicability in building automated diagnostic system. Liang et al. (2016) presented a 16-layer CNN towards classifying uninfected and parasitized cells. The study reported that the custom model was more accurate, sensitive, and specific than the pre-trained model. Dong et al. (2017) evaluated three well-known CNNs (LeNet, AlexNet and GoogLeNet) on classifying parasite/not parasite slide images of thin blood stains. Simulation results showed that all three CNNs achieved classification accuracy scores of over $95\%$. Rajaraman et al. (2018b) introduced a pretrained CNN as a feature extractor towards improved malaria parasite detection in thin blood

smear images, and the results present the use of pretrained CNNs as a promising tool in malaria detection.

Rajaraman et al. (2018a) has demonstrated that deep neural networks can be used to detect malaria from microscopic images. However, in order to deploy such systems in medical facilities, it is vital to ensure that the automated detection systems are indeed robust. Nonetheless, deep learning algorithms work on the premise that both training and test data are drawn from the same application-specific distribution. However, in real-world applications, they need to be able to robustly handle anomalous inputs including, 1) adversarial samples arising from image distortion and 2) samples drawn from a different distribution but belong to the same input space.

In this work, we propose using a Mahalanobis distance-based confidence score method (Lee et al., 2018; Kamoi & Kobayashi, 2020; Nitsch et al., 2021) for detecting abnormal (both OOD and adversarial) samples to improve the performance and robustness of pre-trained convolutional neural network models to improve the robustness of malaria detection (Figure 1). The suggested method gives better results compared to the current state-of-the-art method ODIN (Liang et al., 2018) in detecting OOD malaria samples. We also demonstrate that Mahalanobis distance-based confidence score outperforms the state-of-the-art detection, LID, in all test cases, in detecting adversarial samples generated by four adversarial attacking methods: FGSM (Goodfellow et al., 2015), BIM (Kurakin et al., 2016), DeepFool (Moosavi-Dezfooli et al., 2016), and CW (Carlini & Wagner, 2017).

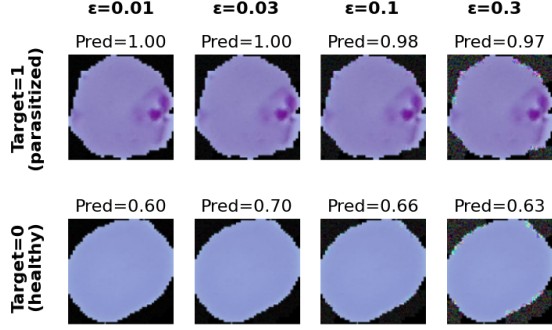

*Figure 1.* FGSM adversarial attack on parasitized and healthy cells for varying levels of noise. Noise can be visually inspected on the black areas and cell boundary, especially for $\epsilon = 0.3$). On the top row, parasites can be seen in dark color inside the cell. The prediction probabilities are indicated above each image.

## 2. Robustness of deep learning models

The robustness of deep learning algorithms needs to be evaluated before deploying them in real-wold. Therefore, it is crucial to ensure the neural networks can detect abnormal inputs in safety- and security-sensitive applications such as medical diagnosis, biometric authentication, intrusion detection, and autonomous driving (Emmott et al., 2016; Nitsch et al., 2021).

### 2.1. Robust out-of-distribution detection for neural networks

Out-of-Distribution (OOD) samples, the test samples that are not well covered by training data, is a major cause of poor performance in deep learning models. OOD samples are able to both evade the deep learning algorithms as well as achieve targeted misclassification with high confidence. There are currently many approaches that can detect OOD examples. They work well when tested on natural samples from a distribution that is sufficiently different from the distribution of the training data (Chen et al., 2020).

Hendrycks & Gimpel (2016) recently proposed a baseline for detecting misclassified and OOD examples in deep neural networks (DNNs), and Liang et al. (2017) improved it by processing the input and output of the DNNs. The Softmax Baseline Mode computes softmax probabilities with the fast-growing exponential function. Thus minor changes to the softmax inputs, can lead to major changes in the output distribution. A softmax baseline method uses probabilities from softmax distributions to predict whether a test example is from a different distribution from the training data or from within the same distribution. Liang et al. (2018) proposed ODIN (Out-of-DIstribution detector for Neural networks) which is a simple and effective method for detecting OOD images in neural networks. ODIN does not require re-training the neural network and is compatible with diverse network architectures and datasets.

### 2.2. Robust adversarial detection for neural networks

Recent studies have concentrated on identifying adversarial examples despite the inefficiency of adversarial defense (Fawzi et al., 2016). Goodfellow et al. (2014) suggested a framework for estimating abnormal samples via adversarial networks. Local Intrinsic Dimensionality (LID) is one of the successful adversarial detection techniques proposed by Ma et al. (2018). With the assumption that adversarial subspaces are low probability regions that are densely scattered in the high dimensional representation space of DNNs. The properties of adversarial regions is considered as a key requirement for adversarial defense (Ma et al., 2018).

## 3. Mahalanobis confidence score

One of the limitations of both ODIN and LID is that they are designed for either OOD or adversarial corruption but not for both. More recently, Lee et al. (2018) proposed a simple yet effective method for detecting both OOD samples and adversarial samples.

Mahalanobis distance-based confidence score is a class-conditional anomaly detection method, motivated by classification prediction confidence (Kamoi & Kobayashi, 2020).

Let us consider a dataset $\mathcal{D} = \{(\mathbf{x}_n, y_n)\}_{n=1}^N$ with input-label pairs. The labels belong to one of the classes $\{1, \ldots, C\}$. For malara parasite detection, labels are either parasitized or healthy. For a deep neural network, $f_\phi$, with parameters $\phi$, we consider a pre-trained softmax classifier,

$$p_\theta(y = c|\mathbf{x}) = \frac{\exp(\mathbf{w}_c^\top f_\phi(\mathbf{x}))}{\sum_{c'} \exp(\mathbf{w}_{c'}^\top f_\phi(\mathbf{x}))}. \quad (1)$$

For each class, we define a multivariate Gaussian distribution, $p(f_\phi(\mathbf{x})|y = c) = \mathcal{N}(f_\phi(\mathbf{x})|\mu_c, \Sigma)$ with class mean $\mu_c$ and pooled-covariance $\Sigma$. This way, we compute the empirical statistics $(\hat{\mu}_1, \hat{\mu}_2, \ldots, \hat{\mu}_C, \hat{\Sigma})$ from the training dataset $\mathcal{D}$. With these statistics, the closest class $\tilde{c}$ to a query input $\mathbf{x}_*$ can be computed using the Mahalanobis distance,

$$\tilde{c} = \min_{c \in \{1,2,\ldots,C\}} \sqrt{(f(\mathbf{x}_*) - \hat{\mu}_c)^\top \hat{\Sigma}^{-1} (f(\mathbf{x}_*) - \hat{\mu}_c)}. \quad (2)$$

Following Liang et al. (2017), a controlled noise $\epsilon$ is added to the input,

$$\tilde{\mathbf{x}}_* = \mathbf{x}_* - \epsilon \cdot \text{sign}\big(\nabla_{\mathbf{x}} (f(\mathbf{x}_*) - \mu_{\tilde{c}})^\top \hat{\Sigma}^{-1} (f(\mathbf{x}_*) - \hat{\mu}_{\tilde{c}})\big), \quad (3)$$

for better calibration. Then, we can compute the confidence score,

$$M(\tilde{\mathbf{x}}_*) = \max_{c \in \{1,2,\ldots,C\}} -(f(\tilde{\mathbf{x}}_*) - \hat{\mu}_c)^\top \hat{\Sigma}^{-1} (f(\tilde{\mathbf{x}}_*) - \hat{\mu}_c). \quad (4)$$

By doing this for all $l = \{1, \ldots, L\}$ layers of the neural network with weights $\alpha_l$ of the logistic regression classifier (separately trained for each layer on a validation dataset (Ma et al., 2018)), we can compute the overall score $M^*(\mathbf{x}_*) = \sum_{l=1}^L \alpha_l M_l(\mathbf{x}_*)$. For a given threshold $\rho$, the query samples, $\mathbf{x}_*$, is *in-distribution*, if $M^*(\mathbf{x}_*) \geq \rho$.

## 4. Experiments

In our experiments, we used a publicly accessible and annotated malaria dataset of healthy and infected blood smear images [1]. It contains 13,779 parasitized and 13,779 uninfected cell images. We split the dataset into $60:10:30$ for

train, validation, and test datasets, respectively. We resized the images to $125 \times 125$ pixels and normalized them to assist in faster convergence. To prevent over-fitting and to account for possible variations in photomicroscopy, we have applied data augmentation techniques such as rotation, shearing, translation, and zooming. For OOD, another malaria dataset consisting of 22,046 was used [2].

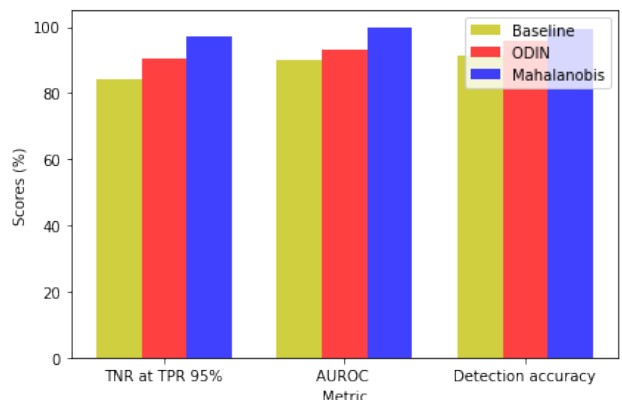

*Figure 2.* Robustness against *out-of-distribution* samples: ResNet-18

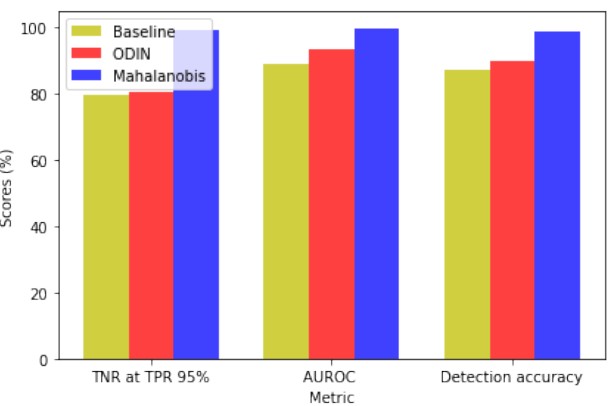

*Figure 3.* Robustness against *out-of-distribution* samples: VGG-19

For OOD and adversarial samples detection, the suggested method to improve the robustness of DL models was evaluated on both VGG-19 and ResNet-18 using a threshold-based detector. We evaluate the models with the following metrics: the true negative rate (TNR) at 95% true positive rate (TPR), the area under the receiver operating characteristic (AUROC) curve, the area under the precision-recall (AUPR) curve, and the detection accuracy. The Mahalanobis

---

[1] https://lhncbc.nlm.nih.gov/publication/pub9932

[2] https://github.com/shriyakabra97/malaria-parasite-detection

*Table 1.* Robustness against *adversarial samples*: a comparison of the performance of LID and Mahalanobis (proposed) towards detecting adversarial test samples generated from malaria image datasets.

| Model | Metric | LID | | | | Mahalanobis | | | |
|---|---|---|---|---|---|---|---|---|---|
| | | FGSM | BIM | DeepFool | CW | FGSM | BIM | DeepFool | CW |
| **VGG-19** | TNR at TPR 95% | 99.96 | 96.87 | 74.48 | 75.93 | 100.00 | 100.00 | 75.96 | 98.11 |
| | AUROC | 97.30 | 96.68 | 78.01 | 89.64 | 99.98 | 99.68 | 83.56 | 99.35 |
| | Detection accuracy | 99.41 | 90.46 | 46.05 | 72.96 | 99.98 | 99.99 | 61.95 | 97.51 |
| **ResNet-18** | TNR at TPR 95% | 96.84 | 95.61 | 63.59 | 73.09 | 99.99 | 97.98 | 76.22 | 98.90 |
| | AUROC | 97.30 | 96.68 | 78.01 | 89.64 | 99.98 | 99.68 | 83.56 | 99.35 |
| | Detection accuracy | 97.02 | 97.65 | 49.56 | 84.66 | 99.75 | 99.95 | 64.95 | 98.05 |

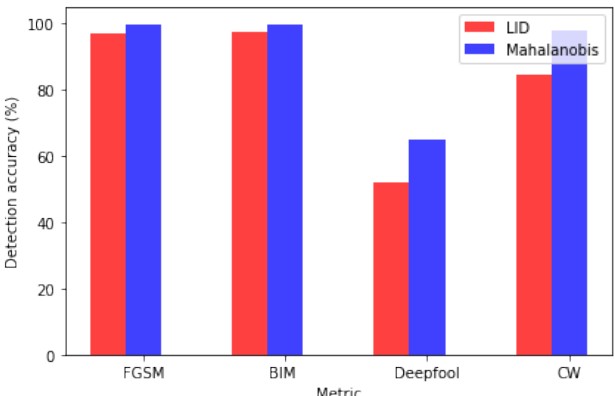

*Figure 4.* A comparison of performance between LID and Mahalanobis distance-based confidence score for ResNet-18 pre-trained model.

*Figure 5.* A comparison of detection performance between LID and Mahalanobis distance-based confidence score for VGG-19 pre-trained model

confidence score was compared with the baseline method and state-of-the-art ODIN for OOD samples. It was also compared with the state-of-the-art LID toward adversarial samples detection. Comparing with the baseline method, ODIN and LID, as shown in Table 1 and Figures 2, 3,4,and 5, the proposed approach outperforms on detecting abnormal samples.

As the Mahalanobis distance-based score method outperforms for the tasks of detecting OOD samples and adversarial samples, it can serve as a diagnostic framework for evaluating deep neural networks, as it is able to reveal their potentially non-obvious vulnerabilities and reliability. Such frameworks help to ensure that deep neural networks are effective, secure, and easy to deploy in a broad range of medical imaging applications beyond malaria detection. Our future work will extend this framework to test medical images under various lighting and other possible sources of corruption. We envision this, in the long-term, will enable low-cost, yet reliable, imaging procedures.

## Broader impact statement

Broadly, our research is a step towards developing robust semi-automated medical image analysis techniques. Specifically, we focus on ensuring that malaria detection procedures are reliable enough before deployment. It, in the long-term, will help low-income countries to eradicate malaria. These systems, however, need to be rigorously validated before deploying in medical facilities.

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
