# OpenReview forum: "Out of Distribution Detection and Adversarial Attacks on Deep Neural Networks for Robust Medical Image Analysis"
_ICML.cc/2021/Workshop/AML — ICML 2021 Workshop AML Poster_

### Official Review · Reviewer_LDfS · 2021-06-20
**Interesting work**

**Rating:** Accept
**Confidence:** 5

**Review:**

This work proposes to use Mahalanobis scores to detect OOD and adversarial samples in medical image analysis, which can benefit more people to eradicate malaria. The contents could be more intriguing if the authors can better motivate on why adversarial robustness is important in medical image analysis, i.e., how an adversary is motivated to attack a malaria analysis system. This clarification can bring more attention to the related research to build adversarially robust medical image analyses.

---

### Decision · Program_Chairs · 2021-06-21

**Decision:**

Accept (Poster)

**Comment:**

This paper proposed to used Mahalanobis scores to detect OOD and adversarial samples in medical image analysis. The reviewer suggested to discuss why adversarial robustness is important in medical image analysis, which make this paper well-motivated for related research.